# Identification of African Swine Fever Virus Inhibitors through High Performance Virtual Screening Using Machine Learning

**DOI:** 10.3390/ijms222413414

**Published:** 2021-12-14

**Authors:** Jiwon Choi, Dongseob Tark, Yun-Sook Lim, Soon B. Hwang

**Affiliations:** 1College of Pharmacy, Dongduk Women’s University, Seoul 02748, Korea; 2Laboratory for Infectious Disease Prevention, Korea Zoonosis Research Institute, Jeonbuk National University, Iksan 54596, Korea; tarkds@jbnu.ac.kr; 3Laboratory of RNA Viral Diseases, Korea Zoonosis Research Institute, Jeonbuk National University, Iksan 54596, Korea; sbhwang@jbnu.ac.kr; 4Ilsong Institute of Life Science, Hallym University, Seoul 07247, Korea

**Keywords:** African swine fever virus, cangrelor, fostamatinib, molecular docking, *Asfv*PolX, machine learning

## Abstract

African swine fever virus (ASFV) is a highly contagious virus that causes severe hemorrhagic viral disease resulting in high mortality in domestic and wild pigs, until few antiviral agents can inhibit ASFV infections. Thus, new anti-ASFV drugs need to be urgently identified. Recently, we identified pentagastrin as a potential antiviral drug against ASFVs using molecular docking and machine learning models. However, the scoring functions are easily influenced by properties of protein pockets, resulting in a scoring bias. Here, we employed the 5′-P binding pocket of *Asfv*PolX as a potential binding site to identify antiviral drugs and classified 13 *Asfv*PolX structures into three classes based on pocket parameters calculated by the SiteMap module. We then applied principal component analysis to eliminate this scoring bias, which was effective in making the SP Glide score more balanced between 13 *Asfv*PolX structures in the dataset. As a result, we identified cangrelor and fostamatinib as potential antiviral drugs against ASFVs. Furthermore, the classification of the pocket properties of *Asfv*PolX protein can provide an alternative approach to identify novel antiviral drugs by optimizing the scoring function of the docking programs. Here, we report a machine learning-based novel approach to generate high binding affinity compounds that are individually matched to the available classification of the pocket properties of *Asfv*PolX protein.

## 1. Introduction

African swine fever virus (ASFV) is an enveloped, double-stranded, 170–193 kbp DNA virus belonging to the *Asfarviridae* family, genus Asfivirus, which replicates predominantly in the cytoplasm of macrophages [1]. The virus has a complex structure and a genome that mainly replicates in the cytoplasm of infected cells [2]. The virus causes a high mortality rate and is responsible for serious economic and production losses worldwide [3,4]. Recently, ASFV outbreaks that have occurred in EU countries and in Transcaucasian, especially in East Asian countries, have resulted in the culling of over one million pigs [5,6,7]. Vaccine and antiviral drugs are the best defenses against the spread of infection, but efforts to develop effective vaccines against ASFVs have failed [8]. Currently, no vaccine or treatment is available for ASFV infection, but recent attempts to generate DNA vaccines, vectored vaccines, or live-attenuated vaccines by defined gene deletions have yielded encouraging data in in vitro assays [9,10,11,12,13]. However, the important issue regarding the safety of live-attenuated ASFV vaccines still exists, and their implementation is highly limited. Several studies have attempted to identify multiple compounds that can inhibit ASFV infections. Antiviral drugs that have identified targets and known mechanisms are classified into five distinct categories: nucleoside analogs, interferons (IFNs), plant-derived compounds, antibiotics, small interfering RNA, and CRISPR/Cas9 [14,15,16,17,18,19,20,21,22,23]. Other antiviral drugs, such as apigenin, resveratrol, and oxyresveratrol, possess potent, dose-dependent anti-ASFV activity in vitro, but they have unknown targets and unknown mechanisms. However, until now, there have been no effective drugs for clinical trials against ASFVs, and there is an urgent need to develop preventive and therapeutic reagents. 

We identified pentagastrin as a potential antiviral drug against ASFVs using principal component analysis and k-means clustering based on molecular docking from previous studies. To identify new potential antiviral drugs, we focused on ASFV DNA polymerase X (*Asfv*PolX), which helps synthesize DNA as an object of ASFV recovery enzyme regulation and virus prevention, as in a previous study. *Asfv*PolX plays an essential role in the DNA repair process of the ASFV genome, and viral replication partially depends on the function of *Asfv*PolX. In addition, *Asfv*PolX has several unique structural features, including a 5’-P binding pocket, a His115-Arg127 platform, and hydrophobic residues Val120 and Leu123, which can all affect the catalytic efficiency of *Asfv*PolX. It is expected that blocking the binding pocket with antiviral compounds can inhibit *Asfv*PolX activity and impair the DNA repair process of the viral genome [24]. We also identified that the *Asfv*PolX-DNA complex structure is a suitable template for determining compounds with high binding affinity and can provide important guidance for the discovery of potential antiviral inhibitors for *Asfv*PolX from previous studies.

Therefore, we used the 13 crystal structures of *Asfv*PolX in complex with DNA enzyme to identify potential binding sites and classified the selected binding sites into three classes based on the SiteMap parameter in the present study. After analyzing the differences in pocket properties of the *Asfv*PolX protein structures, we found that large contact areas, large enclosures, and highly hydrophilic pockets could be a significant cause of the score bias for molecular docking. In this study, we performed docking of the 1732 DrugBank dataset to 13 structures of *Asfv*PolX using Glide docking programs and found that some of the 13 structures were highly scored frequency hits resulting from the scoring bias. Scoring functions are used to select the most promising compounds; therefore, the accuracy of the scoring function at this stage is of paramount importance and represents the primary determinant of success or failure in virtual screening [25]. However, the scoring functions are easily influenced by the properties of protein pockets, resulting in scoring bias for proteins with certain properties. To overcome these limitations of virtual screening, we then applied principal component analysis based on 46 molecular descriptors calculated by QikProp to eliminate this scoring bias, which was effective in making the SP Glide score more balanced between 13 *Asfv*PolX structures in the molecular docking of the dataset. These approaches identified cangrelor and fostamatinib as new antiviral drugs. Furthermore, the classification of the pocket properties of *Asfv*PolX protein causing scoring bias against certain structures found in this study can provide an alternative approach to identify novel antiviral drugs by optimizing the scoring function of the docking programs. Thus, these approaches could help improve the accuracy rate of virtual screening for various protein–ligand complexes.

## 2. Results

### 2.1. Classification of Binding Sites for the AsfvPolX Protein

To identify new antiviral drugs targeting the *Asfv*PolX protein–DNA interfaces, we compiled a list of the 13 available crystal structures of *Asfv*PolX in complex with the DNA enzyme identified from the PDB database. Detailed information on the selected structures is provided in Appendix A. We also identified the potential binding sites identified by SiteMap on the DNA complex structures of *Asfv*PolX (Appendix A). Among the predicted binding sites identified by SiteMap, we finally selected a site proximal to the *Asfv*PolX protein–DNA interface as a potential binding pocket to identify new antiviral drugs. The binding pocket predicted by SiteMap was given a Dscore based on pocket parameters, such as size, exposure to solvent, enclosure by protein, ratio of hydrogen bond donors and acceptors, and hydrophobic/hydrophilic ratio. When SiteMap was applied to 13 *Asfv*PolX protein crystal structures, the obtained Dscore of the selected binding site was between 0.802 and 0.479, with an average of 0.637 (Appendix A). To explore the characteristics of the selected binding sites of each *Asfv*PolX protein structure, we performed hierarchical clustering analysis and principal component analysis (PCA). Clustering of the selected pockets represented a clear separation into three classes on the SiteMap parameters of binding pockets. Thus, the identified pockets can be classified into three classes as follows: Class I contains relatively higher exposure pockets, Class II contains generally higher enclosure pockets, and Class III contains higher hydrophilic pockets (Figure 1A,B). In particular, most structures in Class I appeared to contain high-exposure pockets. Of these, the target structures belonging to Class II and Class III generally exhibited higher hydrophilicity, enclosure, and contact with proteins; these structures can be considered challenging drug targets. These features are more easily defined by probability density curves for SiteMap parameters with high relevance of binding pockets for the three classes (Figure 2). These clustering approaches allowed us to obtain information regarding trends in the druggability and properties of the binding sites, and, also, confirm the criteria for selecting the representative conformations for molecular docking. 

### 2.2. Druggability Analysis of the Binding Sites

To investigate the distribution of the docking score for the three classes, we conducted docking simulation using Glide software (Schrödinger, New York, NY, USA). First, molecular docking was performed using 13 different types of *Asfv*PolX protein structures (classified into three classes) using FDA-approved drugs from the DrugBank database. After performing molecular docking, the dataset for each of the protein structures contained approximately 1712–1750 compounds with docking scores. Figure 3A, B shows the distribution of the docking scores for each *Asfv*PolX structure. By comparing the distribution of the docking scores, we observed that the docking scores of some structures had a different range. The structures with the largest difference in docking score distributions were 5HRG, 5HRH, 5HRK, and 5HRL (PDB ID). Except for 5HRG, all three other structures belonged to Class III. These results indicate that scoring functions may have different degrees of scoring bias for various structures on the same protein, and some structures result in a high overall docking score. Based on the previous clustering results, the four structures confirmed that although they were not included in the same class, they had highly common protein pocket properties, such as higher hydrophilic character, degree of enclosure, and degree of contact (Figure 2). As shown in Appendix A, the hydrophilic properties are strongly correlated with the enclosure and contacts among the eight pocket parameters. Thus, some structures belonging to Class II or Class III with such pocket characteristics on the *Asfv*PolX protein surface can be classified as suitable templates for virtual screening. The four structures (5HRG, 5HRH, 5HRK, and 5HRL) have a much greater range of a high docking score (close to −10) distributions, which typically demonstrated the common pocket properties of higher hydrophilicity, enclosure, and contact by protein. These trends prove that the main properties for classifying the druggable binding pocket on the *Asfv*PolX protein surface to find antiviral compounds with high binding affinity are the hydrophilic character, degree of enclosure, and degree of contact.

### 2.3. Comparison of Docking Scores between Three Classes 

As previously mentioned, we observed a scoring bias for 13 *Asfv*PolX crystal structures in molecular docking. To identify the docking score distributions, docking results were analyzed based on the three classes according to the pocket properties classified in previous studies. The docking score distribution of the three classes is shown in Figure 4. The mean docking scores of compounds in each of the three classes were −6.253, −6.616, and −8.282, respectively, indicating that the target structures for Class III could predict compounds with higher binding affinity than the target structures for Class I and Class II (Figure 4A). As shown in Figure 3B, the docking results of the 1712 DrugBank dataset to 13 crystal structures of *Asfv*PolX protein found that highly scored hits mostly belonged to Class III resulting from the scoring bias. Therefore, the properties of protein pockets causing scoring bias for specific target structures we found here can provide a theoretical basis for further optimization of docking results for further research. Taken together, these results indicate that the structures belonging to Class III are the best templates for predicting compounds with high binding affinity and can also provide important guidance for designing potential *Asfv*PolX antiviral inhibitors.

### 2.4. PCA-Based Clustering on the Three Classes of Dataset

To eliminate this scoring bias that results in numerous false positives in molecular docking, we employed an effective approach. We generated a dataset based on docking scores for *Asfv*PolX protein structures of compounds obtained from the DrugBank database. First, 100 compounds with the best docking scores for each structure were selected, and a total of 1300 compounds was obtained (see Methods section for additional information); then, 200 compounds with the best docking scores for each class were selected, ultimately resulting in a total of 600 compounds. The docking score distribution for this dataset is shown in Appendix A. Next, to identify different patterns in the chemical space for the three classes within the dataset, 46 descriptors calculated from the QikProp module of each molecule were calculated and applied for PCA. PCA is an unsupervised learning method that simplifies the complexity of high-dimensional datasets while maintaining trends and patterns. In Figure 5A, we present a two-dimensional score plot, which shows how the three classes were distributed on PCs. The PCA-based visualization confirmed that most of the compounds within the dataset shared a chemical space between the three classes. Notably, three classes showed that a centralized distribution was clearly observed along with a prominent pattern with vast chemical diversity. Figure 5B shows the loading plot vectors for each physicochemical property. Then, we employed *k*-means clustering on the PCA approach to obtain a better cluster solution (Figure 5C). We expected that the chemical diversity region of the dataset via PCA could provide important chemical regions for identifying compounds with higher binding affinity to *Asfv*PolX proteins than centralized regions for the three classes. For the datasets, we identified the optimal clustering consisting of groups of two clusters. The docking score distribution of the two clusters is shown in Figure 5D, which demonstrates that the docking score of clusters has a different range. For compounds in Clusters 1 and 2, the mean docking scores were −6.85 and −7.0, respectively. The docking score distribution in Figure 5D shows that Cluster 2 with a chemical diversity region includes more compounds with a docking core belonging to the −10 to −9 range than Cluster 1. These results showed that 46 physicochemical descriptor-based clustering can accurately provide a unique region containing compounds with high affinity in the dataset. 

### 2.5. Selection of Candidate Compounds

The compounds within Cluster 2 were clustered together in the circular hierarchical cluster analysis (HCA) dendrogram and four clusters were identified, as shown in Figure 6. The compounds within the same cluster will be chemically similar, so that some neighboring compounds selected from a cluster might be expected to exhibit similar behavior. This dendrogram indicates that cangrelor, fosaprepitant, and fostamatinib share structural similarities and exist in the same cluster when classified into the three clusters, such as pentagastrin, which has been identified as an ASFV inhibitor in previous studies (Appendix A) [26]. These data imply that these compounds exhibit similar behavior, and thus were selected for further analysis.

### 2.6. Cangrelor and Fostamatinib Inhibited AsfvPolX Activity

To measure the effect of cangrelor, fosaprepitant, and fostamatinib on *Asfv*PolX polymerase activity, BL21 (DE3)-RIL cells carrying the pET28-*Asfv*PolX plasmid were cultured and the *Asfv*PolX protein was purified using a Ni-NTA column and further purified using a glycerol gradient [26]. *Asfv*PolX (100 ng) was used for the polymerase assay to analyze the effect of the three compounds on *Asfv*PolX polymerase activity. As shown in Figure 7, *Asfv*PolX activity was significantly decreased by cangrelor and fostamatinib in a dose-dependent manner. However, fosaprepitant showed little or no reduction in *Asfv*PolX activity (Appendix A). These data suggest that cangrelor and fostamatinib may be potential drug candidates for African swine fever virus.

## 3. Discussion

The current ASFV epidemic could have serious global repercussions for food security and economic stability. To date, there are no vaccines or specific drugs against ASFV drugs. Therefore, there is an urgent need to find effective drugs to identify new anti-ASFV drugs. We recently reported molecular docking-driven machine learning models to predict novel antiviral drugs against ASFVs and predicted 10 top-ranked compounds with high docking scores using PCA and *k*-means clustering. These methods identified pentagastrin as a potential antiviral drug against ASFVs, and, also, observed that the compound had an inhibitory effect on *Asfv*PolX activity. Virtual screening is a promising method for obtaining novel hit compounds in drug discovery from molecular docking experiments and predicts whether and how small molecules bind to a macromolecular target using a suitable 3D structure. Scoring functions for structure-based virtual screening are primarily used for pose and affinity prediction. However, the scoring functions used to approximately predict the binding affinity would be easily influenced by the properties of protein pockets, resulting in scoring bias for proteins with certain properties. Thus, accurate prediction of binding affinity remains a challenging task and is critical to the success of structure-based virtual screening. The identification of promising lead compounds through a large compound library involves significant efforts to develop more sophisticated and accurate models. Machine learning is a major subfield of artificial intelligence for extracting meaningful information from large-scale data. In this study, we focused on cluster analysis and PCA in machine learning to make docking scores more balanced between different structures of the dataset to accurately filter potential lead compounds from virtual screening. The classification of pocket properties of *Asfv*PolX proteins can provide an alternative approach for identifying new antiviral agents by optimizing the scoring capabilities of molecular docking that cause scoring bias for specific structures. First, we analyzed the properties of binding pockets by clustering the selected binding sites of the *Asfv*PolX protein to select the representative conformations for virtual screening. In this step, we detected binding pockets in each conformation using the SiteMap module and calculated the properties of binding pockets, such as size, volume, degree of enclosure/exposure, degree of contact, hydrophobic/hydrophilic character, hydrophobic/hydrophilic balance, and hydrogen-binding possibility (acceptors/donors). Next, we performed molecular docking of the 1712 DrugBank dataset to 13 crystal structures of *Asfv*PolX protein using the Glide docking program. From these results, we found that highly scored hits mostly belonged to Class III resulting from the scoring bias. The median average rank of docking scores increases in Class III, which means that the target structures in Class III have a better average rank than other target structures. To eliminate this scoring bias that results in many false positives in molecular docking, we applied PCA and clustering to make the SP Glide score more balanced between 13 *Asfv*PolX structures in the dataset. This machine learning model is a promising new tool for understanding and interpreting the bias of scoring functions. Therefore, we identified three compounds that exist in clusters, such as pentagastrin, and share structural similarities, which were considered for further studies. These studies have led to the identification of two candidate compounds for which in vitro antiviral activity could be shown in a biological assay.

In conclusion, machine learning models applied in virtual screening could significantly increase the accuracy of docking by Glide. These approaches will help predict the true target structure and improve the limitations of docking programs with scoring bias for these non-target structures. The present study, performed through molecular docking-driven machine learning models, can play an important role in identifying potential antiviral drugs against ASFVs.

## 4. Materials and Methods

### 4.1. Protein Structure Selection and Preparation

The *Asfv*PolX protein structures with UniProt ID (P42494) were retrieved from the PDB database (https://www.rcsb.org, accessed on 26 April 2021), and the protein structures obtained by NMR spectroscopy were removed. The dataset of 13 *Asfv*PolX protein structures obtained by X-ray diffraction was selected for further studies. Detailed information on the selected structures is provided in Appendix A. The crystal structures were imported into Maestro (Schrödinger, LLC, New York, NY, USA, 2020), and, then, prepared using the Protein Preparation Wizard. 

### 4.2. Identification of Druggable Pockets

To characterize the putative binding sites, the prepared protein structures were submitted to the SiteMap module as implemented in Schrödinger Suite 2020 [27,28]. The SiteMap module can be used to identify and clarify potential binding sites (site points) that most likely contribute to tight protein–ligand or protein–protein binding. The following physicochemical properties of the binding sites were calculated using the SiteMap program: size, volume, degree of enclosure/exposure, degree of contact, hydrophobic/hydrophilic character, hydrophobic/hydrophilic balance, and hydrogen-binding possibility (acceptors/donors). The five top-ranked potential binding sites of each of the 13 structures were identified, and the parameters of the surface pockets close to the DNA-binding site were selected for clustering analysis. Clustering of the SiteMap parameters was performed using the heatmap library in R [29,30]. The SiteMap parameters were transformed using “percentize”, and average linking was employed for hierarchical clustering. Data were log-transformed to normalize the SiteMap parameters, and clusters were generated using the group medians and hierarchical clustering with an uncentered Pearson correlation to generate a complete linkage to create the most unbiased set of clusters.

### 4.3. Virtual Screening (Docking Simulation)

To examine the binding interactions of *Asfv*PolX-ligand complexes, molecular docking studies were performed using Glide software (Schrödinger, New York, NY, USA), which uses an optimized potential for liquid simulations (OPLS)-2005 force field. A test set of 13 *Asfv*PolX crystal structures containing DNA-binding domains was used to assess the binding affinity. The molecules tested in this study consisted of 2413 FDA-approved drugs obtained from the DrugBank database (https://go.drugbank.com; DrugBank Release Version 5.1.4, accessed on 2 July 2019). Prior to screening, the database was filtered to select compounds with molecular weights >200 Da, and 1732 drugs were obtained. LigPrep (Schrödinger, New York, NY, USA) was used to generate 3D ligand structures. The active grid was generated using the Receptor grid application in the Glide module. Docking was performed on a defined receptor grid using the standard precision mode of Glide [31,32]. The best docking pose for a compound was selected based on the best scoring conformations from the Glide score. 

### 4.4. Calculation of Molecular Descriptors

After SP docking, the QikProp module of Schrödinger (Maestro, Schrödinger, LLC, New York, NY, USA, 2020) was used to calculate molecular descriptors. A molecular descriptor is a numerical description that represents physical and chemical information of a compound. A set of ADME-related properties (46 molecular descriptors) for the dataset was calculated. It generates physically relevant descriptors and uses them to predict the absorption, distribution, metabolism, and elimination (ADME) properties. 

### 4.5. Machine Learning Models

To analyze PCA-based clustering, the dataset first selected 1300 compounds (200 top-ranked compounds chosen from docking results of over 1712 compounds from the DrugBank database against 13 *Asfv*PolX protein structures), which eventually consisted of a total of 600 compounds, 200 for each of the three classes based on the SiteMap parameters of binding pockets (See Section 2.1). Pearson and Spearman correlation calculations were performed using R package. PCA is a multivariate statistical method used for exploratory data analysis. It allows representation of the property space by projection into the principal component plane (PC1 and PC2), which is the mathematical equivalent of taking a picture from the most favorable angle [33,34]. To extract the most important information from the dataset, PCA was employed to explore the chemical space of antiviral compounds as a function of the 46 molecular descriptors using the FactoMineR package in R. PCA clustering was performed for each dataset using the k-means clustering within the Factoextra R package. All workflows and procedures were generated using the R software version 4.0.1 (6 June 2021).

### 4.6. Measurement of Polymerase Activity

The pET28-PolX plasmid was transformed into *Escherichia coli* BL21 (DE3)-RIL. PolX was purified using a Ni-NTA column (Qiagen, Valencia, CA, USA) and a HiTrap SP HP column (GE Healthcare, Chicago, IL, USA) [26]. The enzyme activity of PolX was measured using the polymerase activity assay kit (Enzynomics, Daejeon, Korea) according to the manufacturer’s instructions. Briefly, PolX protein was mixed with cangrelor, fostamatinib, and fosaprepitant (Sigma-Aldrich, St. Louis, MO, USA) for 10 min at 4 °C, and 2X reaction buffer and primed substrate were added to the enzyme mixture. For PolX induction, the mixture was incubated for 30 min at 37 °C. Following inactivation of polymerase activity, the synthesized DNA was denatured at 95 °C for 20 min. For annealing DNA, the temperature was decreased at 65 °C for 10 min, 45 °C for 5 min, and 25 °C for 5 min. To measure the content of synthesized DNA, 100 cycles of 5 s, with 0.5 °C temperature increments from 45 °C to 95 °C, were used for the melting curves using a CFX Connect real-time system (Bio-Rad Laboratories, Hercules, CA, USA).

## Figures and Tables

**Figure 1 ijms-22-13414-f001:**
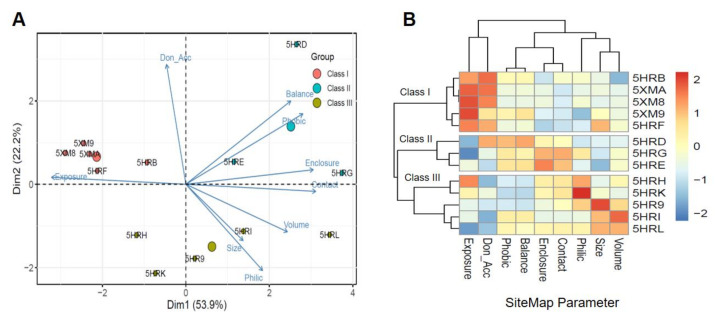
PCA plot and heatmap of selected binding sites on the *Asfv*PolX protein. (**A**) PCA of selected potential binding sites on *Asfv*PolX protein structures. The loading plot vectors are represented by arrows for each SiteMap parameter. The red, blue, and green dots represent Class I, Class II, and Class III, respectively. (**B**) The heatmap clustering of the SiteMap parameters of the selected binding sites on *Asfv*PolX protein. Blue color represents low and red color represents high score levels. Both the heatmap and PCA were constructed using R software version 4.0.1 (6 June 2021).

**Figure 2 ijms-22-13414-f002:**
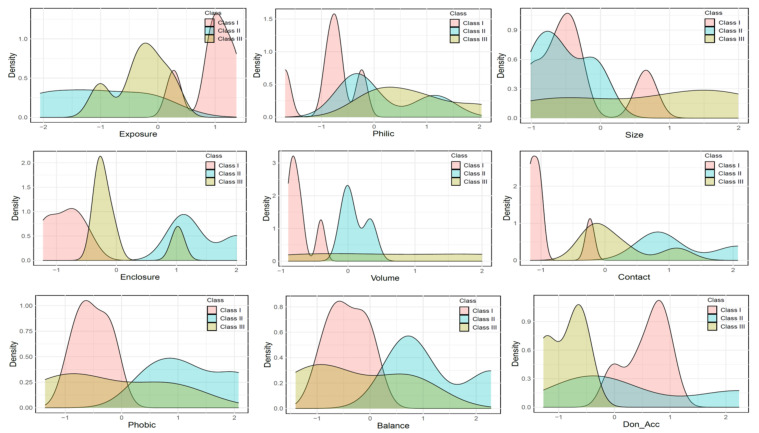
Probability density curves for the highly relevant SiteMap parameters of the binding pocket for three classes. The X-axis represents the SiteMap parameters, and the Y-axis represents the probability density.

**Figure 3 ijms-22-13414-f003:**
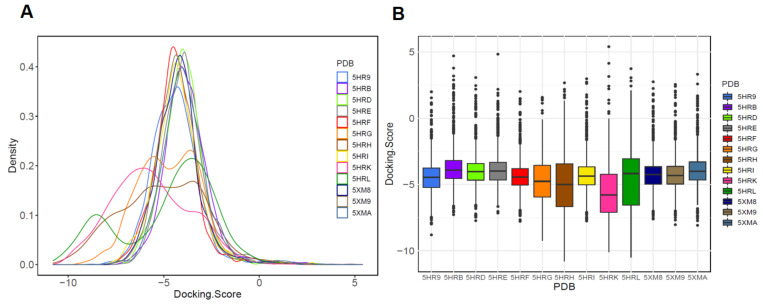
Comparison of docking results obtained from selected binding sites on each *Asfv*PolX protein. (**A**) Probability density curves of the Glide docking score. The X-axis represents the docking score of each structure-ligand pair, and the Y-axis represents the probability density. (**B**) Boxplots for Glide docking score distributions of each structure. The black line indicates the median result for each crystal structure.

**Figure 4 ijms-22-13414-f004:**
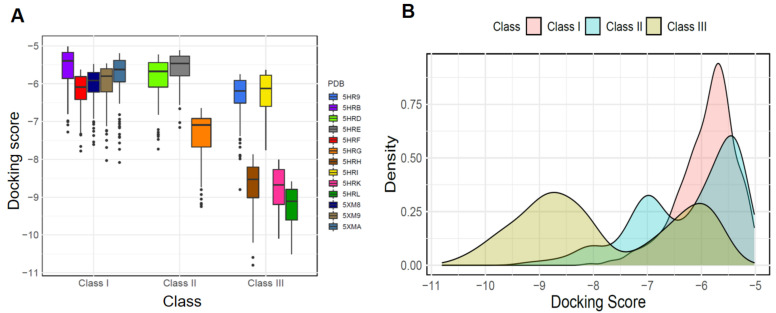
Comparison of the docking performance of three classes across the *Asfv*PolX dataset. (**A**) Boxplots for Glide docking score distributions of each crystal structure. The black line indicates the median result for each crystal structure. (**B**) The probability density curves of Glide docking score. The X-axis represents the docking score of each structure-ligand pair, and the Y-axis represents the probability density.

**Figure 5 ijms-22-13414-f005:**
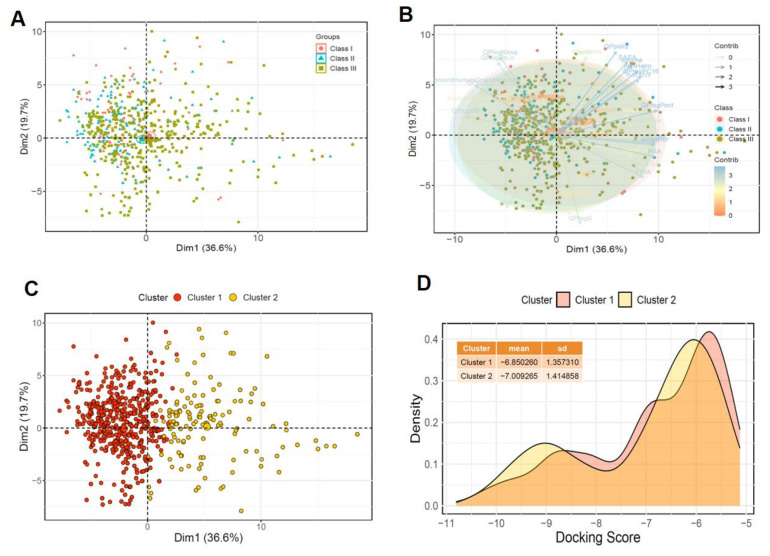
Visual representation of the chemical space and probability density curves of the *Asfv*PolX dataset. (**A**) PCA clustering plot of the chemical space of three classes across the *Asfv*PolX dataset. The red, blue, and green dots correspond to Class I, Class II, and Class III, respectively. (**B**) The loading plot vectors are represented by arrows for each physicochemical property. (**C**) The red and yellow dots correspond to Cluster 1 and Cluster 2, respectively. (**D**) The probability density curves of the Glide docking score. The X-axis represents the docking score of each structure-ligand pair, and the Y-axis represents the probability density.

**Figure 6 ijms-22-13414-f006:**
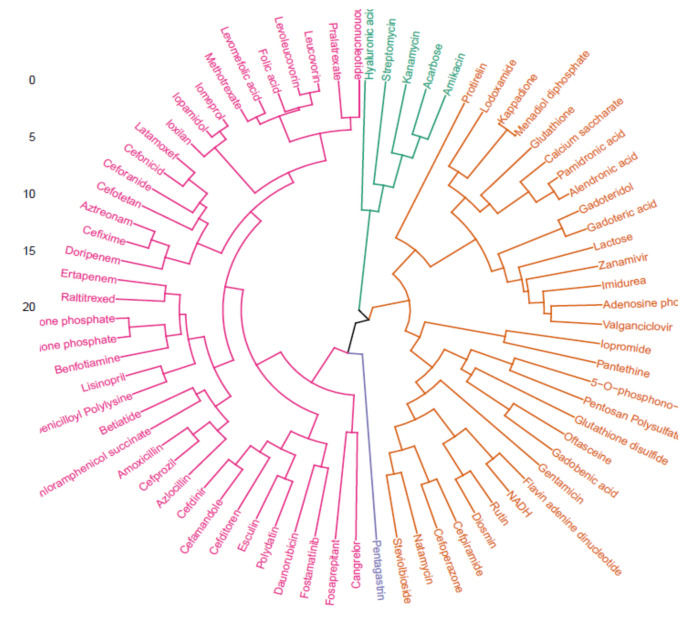
Circular hierarchical cluster analysis (HCA) dendrogram of Cluster II using Euclidean distances and average linkage showing four clusters. Each cluster is colored differently.

**Figure 7 ijms-22-13414-f007:**
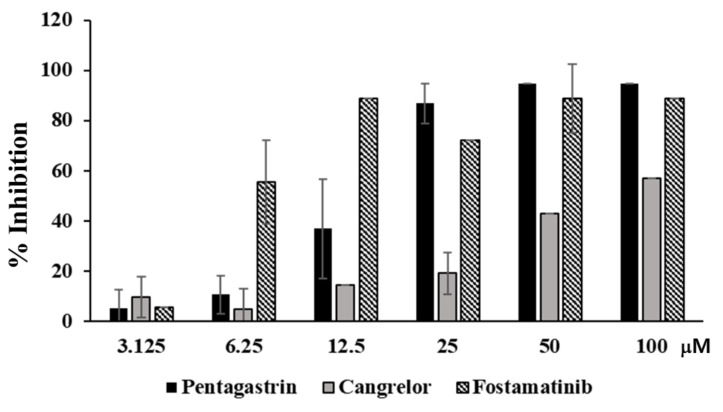
Cangrelor and fostamatinib inhibits *Asfv*PolX activity. Cangrelor and fostamatinib was mixed with 100 ng of *Asfv*PolX for 10 min and polymerase activities were measured as described in Materials and Methods.

## Data Availability

Not applicable.

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
