# Peer review of "Identification of African Swine Fever Virus Inhibitors through High Performance Virtual Screening Using Machine Learning"

_ijms, 2021, doi:10.3390/ijms222413414_

Round 1

Reviewer 1 Report

The authors reported a computational study for the identification of African swine fever virus inhibitors.

The VS study has also been confirmed by experimental assays based on polymerase activity.

There is only one major comment:

In the experimental assay, a reference ligand should be added, in order to provide a direct comparison for the new identified inhibitors.

Reviewer 2 Report

The authors have presented an interesting method to identify inhibitors through machine learning-assisted virtual screening. The manuscript is well written and well presented. The following comments could be addressed for better clarity.

a) The authors should describe their previous work and how it is related to their current work in a more straightforward way in the introduction. I find very few changes in the methodology- except that the previous work looked at only 2 structures and the current one analyzes 13 structures(. In addition to this if there are major changes in the strategy that needs to be explained in the manuscript so that the novelty of the work is ensured. )-Has this extension of work resulted in much potent inhbitors?-that is not very clear from the data.

b) I wonder why did the authors remove the NMR structures from the analysis when the NMR ensembles can be a rich representation of diverse conformations.

c) a one-point polymerase inhibition does not reflect the inhibitory capacity of the inhibitors. The authors should carry out binding studies and kinetic polymerase assays and should correlate the Kd and IC50 with the binding scores to make good sense of this ranking system. Ideally, authors should use Pentagastrin (the inhibitor they found in their earlier work) and compare its effect with the new inhibitors they found in this study to prove the usefulness of this new strategy.

Minor comments :

1)

">To analyze PCA-based clustering, the dataset first selected 1200 compounds (200 top-
>ranked compounds chosen from docking results of over 1712 compounds from the
>DrugBank database against 13 AsfvPolX protein structures) and eventually consisted of a
>total of 600 compounds, 200 for each of the three classes"

Which 3 classes are the authors referring to? I could find it earlier in the text, But if you add a reference here this will make it an easy read.

2)

"> 3.1. Classification of binding sites for the AsfvPolX protein
> 173 To identify new antiviral drugs targeting the AsfvPolX protein–DNA interfaces, we
> first identified potential binding sites using SiteMap module 26,27 , which uses the
> hydrophobicity and accessibility of a detected binding site to assess how likely a small-
>molecule inhibitor is to bind."

About the 13 structures and the Site Identification-it has already been described  in section 2 - Methods and Materials
Section 3 is the results. It does not make sense to mention it in both sections. Maybe this is not a result.

3)
">Taken together, these
>results indicate that the structures belonging to Class III are the best templates for
>determining compounds with high binding affinity and can also provide important
>guidance for designing potential AsfvPolX antiviral inhibitors."

This conclusion might be questionable. At least for real binding affinity. It is only true for the predicted binding affinity. But what matters, in the end, is which structures are physiologically most relevant. 

Round 2

Reviewer 1 Report

It can be accepted for pubblication